# Comparison of Active Surveillance to Stereotactic Radiosurgery for the Management of Patients with an Incidental Frontobasal Meningioma—A Sub-Analysis of the IMPASSE Study

**DOI:** 10.3390/cancers14051300

**Published:** 2022-03-03

**Authors:** Abdurrahman I. Islim, Georgios Mantziaris, Stylianos Pikis, Ching-Jen Chen, Adomas Bunevicius, Selçuk Peker, Yavuz Samanci, Ahmed M. Nabeel, Wael A. Reda, Sameh R. Tawadros, Amr M. N. El-Shehaby, Khaled Abdelkarim, Reem M. Emad, Violaine Delabar, David Mathieu, Cheng-Chia Lee, Huai-Che Yang, Roman Liscak, Jaromir May, Roberto Martinez Alvarez, Nuria Martinez Moreno, Manjul Tripathi, Douglas Kondziolka, Herwin Speckter, Camilo Albert, Greg N. Bowden, Ronald J. Benveniste, Lawrence Dade Lunsford, Jason P. Sheehan, Michael D. Jenkinson

**Affiliations:** 1Department of Neurosurgery, The Walton Centre NHS Foundation Trust, Liverpool L9 7LJ, UK; michael.jenkinson@liverpool.ac.uk; 2Institute of Systems, Molecular and Integrative Biology, University of Liverpool, Liverpool L69 7BE, UK; 3Department of Neurological Surgery, University of Virginia, Charlottesville, VA 22903, USA; urf4wf@hscmail.mcc.virginia.edu (G.M.); sp3az@hscmail.mcc.virginia.edu (S.P.); chenjared@gmail.com (C.-J.C.); ab5qs@hscmail.mcc.virginia.edu (A.B.); jps2f@virginia.edu (J.P.S.); 4Department of Neurosurgery, Koc University School of Medicine, Istanbul 34010, Turkey; peker@selcukpeker.com (S.P.); msamanci@kuh.ku.edu.tr (Y.S.); 5Gamma Knife Center Cairo, Nasser Institute, Cairo 11796, Egypt; ahmed_m_nabeel@yahoo.com (A.M.N.); waelareda@hotmail.com (W.A.R.); samehroshdy@yahoo.com (S.R.T.); amrelshehaby@yahoo.com (A.M.N.E.-S.); khalidakm@yahoo.com (K.A.); reememad2004@yahoo.com (R.M.E.); 6Department of Neurosurgery, Benha University, Benha 13512, Egypt; 7Department of Neurosurgery, Ain Shams University, Cairo 11566, Egypt; 8Department of Radiation Oncology, National Cancer Institute, Cairo University, Cairo 12613, Egypt; 9Centre de Recherche du CHUS, Université de Sherbrooke, Sherbrooke, QC J1H 5N4, Canada; violaine.delabar@usherbrooke.ca (V.D.); david.mathieu@usherbrooke.ca (D.M.); 10Department of Neurosurgery, School of Medicine, Neurological Institute, Taipei Veteran General Hospital, Taipei City 11217, Taiwan; yfnaughty@gmail.com (C.-C.L.); wade012@gmail.com (H.-C.Y.); 11Department of Neurosurgery, National Yang-Ming University, Beitou District, Taipei City 11221, Taiwan; 12Department of Radiation and Stereotactic Neurosurgery, Na Homolce Hospital, 150 00 Prague, Czech Republic; roman.liscak@homolka.cz (R.L.); jaromir.hanuska@homolka.cz (J.M.); 13Department of Radiosurgery, Rúber International Hospital, 28034 Madrid, Spain; roberto.martineza.ce@ruberinternacional.es (R.M.A.); nuriamartinez@ruberinternacional.es (N.M.M.); 14Department of Neurosurgery and Radiotherapy, Nehru Hospital Sector 12, Postgraduate Institute of Medical Education and Research, Chandigarh 160012, Punjab, India; drmanjultripathi@gmail.com; 15Department of Neurosurgery, New York University, New York, NY 10016, USA; douglas.kondziolka@nyulangone.org; 16Department of Neurosurgery and Radiation Oncology, New York University, New York, NY 10016, USA; 17Department of Radiology, Dominican Gamma Knife Center and CEDIMAT, Santo Domingo 10514, Dominican Republic; hspeckter@cedimat.net (H.S.); camiloalbertf@gmail.com (C.A.); 18Department of Neurosurgery, 2D1.02 Mackenzie Health Sciences Centre, University of Alberta, Edmonton, AB T6G 2B7, Canada; gregory.bowden@ahs.ca; 19Department of Neurosurgery, University of Miami Miller School of Medicine, Miami, FL 33136, USA; rbenveniste@med.miami.edu; 20Department of Neurosurgery, University of Pittsburgh, Pittsburgh, PA 15260, USA; lunsfordld@upmc.edu

**Keywords:** asymptomatic, incidental, meningioma, surveillance, radiosurgery

## Abstract

**Simple Summary:**

Meningioma, a type of brain tumor, is a common incidental finding on brain imaging. The best management approach for patients with an incidental meningioma remains unclear. This retrospective multi-center study investigated the outcomes of patients with an incidental meningioma in a frontobasal location, who were managed with active surveillance (*n* = 28) compared to stereotactic radiosurgery (SRS) (*n* = 84). Within 5 years of follow-up, SRS improved the radiological control of incidental frontobasal meningiomas (0% vs. 52%), but no symptoms occurred in either group. In the active surveillance cohort, 12% underwent an intervention for tumor growth. The findings of this study provide information to enable shared decision making between clinicians and patients with incidental frontobasal meningiomas.

**Abstract:**

Meningioma is a common incidental finding, and clinical course varies based on anatomical location. The aim of this sub-analysis of the IMPASSE study was to compare the outcomes of patients with an incidental frontobasal meningioma who underwent active surveillance to those who underwent upfront stereotactic radiosurgery (SRS). Data were retrospectively collected from 14 centres. The active surveillance (*n* = 28) and SRS (*n* = 84) cohorts were compared unmatched and matched for age, sex, and duration of follow-up (*n* = 25 each). The study endpoints included tumor progression, new symptom development, and need for further intervention. Tumor progression occurred in 52.0% and 0% of the matched active surveillance and SRS cohorts, respectively (*p* < 0.001). Five patients (6.0%) treated with SRS developed treatment related symptoms compared to none in the active monitoring cohort (*p* = 0.329). No patients in the matched cohorts developed symptoms attributable to treatment. Three patients managed with active surveillance (10.7%, unmatched; 12.0%, matched) underwent an intervention for tumor growth with no persistent side effects after treatment. No patients subject to SRS underwent further treatment. Active monitoring and SRS confer a similarly low risk of symptom development. Upfront treatment with SRS improves imaging-defined tumor control. Active surveillance and SRS are acceptable treatment options for incidental frontobasal meningioma.

## 1. Introduction

The prevalence of incidental findings has increased due to the wider availability of brain magnetic resonance imaging (MRI). Incidental asymptomatic meningiomas are present on 0.9% to 1.0% of the general population’s brain MRIs [1,2]. After discovery of an incidental meningioma, active clinical and MRI surveillance is the recommended first line management strategy until radiological progression or development of neurological signs or symptoms ensue [3]. This is justified by the indolent nature of these tumors. In a prospective study, none of 64 patients with an incidental meningioma recruited became symptomatic over a 5-year duration [4]. Moreover, more than 60% of the meningiomas exhibited a self-limiting growth pattern [4]. In retrospective studies, the risk of rapid exponential meningioma growth was low varying between 7% and 10% [5,6]. Frontobasal meningiomas are frequently non-*NF2* mutated and harbor *TRAF7*, *KLF4*, *AKT1*, and *SMO* genetic alterations, which render their behavior nearly always indolent [7]. Nonetheless, their proximity to critical neurovascular structures, such as the optic pathway, warrants consideration of early intervention before growth and development of symptoms. This approach would avoid excessive meningioma growth that leads to involvement of neurovascular structures and the potential for surgical morbidity and lower rates of gross total resection [8]. Since most incidental meningiomas are smaller than 10 cm^3^ [9], early intervention with stereotactic radiosurgery (SRS) is an alternative management choice. It offers a non-invasive measure for achieving tumor control in 90–100% of patients [10]. However, studies of its efficacy focused primarily on residual frontobasal meningioma and demonstrated a 7–13% risk of adverse events including cranial nerve palsy or cognitive impairment [11,12]. The risk of an adverse event must be weighed against the risk of meningioma growth and development of symptoms. To this end, this sub-analysis of the IMPASSE study [13], an international multi-center comparative study of incidental meningioma progression following active surveillance or SRS, focuses on comparative outcomes of patients with a frontobasal meningioma subject to either early prophylactic intervention with SRS or active surveillance.

## 2. Materials and Methods

### 2.1. Study Design and Setting

IMPASSE was an international, multi-center, retrospective cohort study of patients with an incidental meningioma subject to SRS or active surveillance. The complete study methods have been previously described [13]. In brief, 14 centers in 10 countries submitted data on 1117 patients who were found to have an incidental meningioma. Early SRS was performed in 727 patients, and 388 patients commenced active surveillance. Clinical and radiological outcomes were compared prior to and after matching for baseline variables. The study was managed by the International Radiosurgery Research Foundation (IRRF). Local institutional review board approval was sought prior to sharing the de-identified data with the IRRF coordinating office. This sub-analysis of the IMPASSE study focused on patients with a frontobasal meningioma.

### 2.2. Study Population

Patients with an incidental frontobasal meningioma were included. An frontobasal location included olfactory groove, planum and tuberculum sellae meningiomas. Patients managed with either SRS or active surveillance were included. Meningiomas were defined as extra-axial, dural-based, and homogenously enhancing lesions on contrast enhanced T1-weighted brain MRI with or without dural tail. Patients were excluded from the study if they were <16 years of age, had multiple meningiomas or any symptoms attributable to the meningioma at diagnosis.

### 2.3. Study Procedures

The investigated intervention was SRS at diagnosis. SRS was delivered in a single session using Gamma Knife (Elekta AB, Stockholm, Sweden). Brain MRI and/or CT were used for stereotactic targeting. Radiosurgical planning, using a multi-isocentric approach, and radiation dose were agreed upon by the local multidisciplinary team, which included neurosurgeons, radiation oncologists and medical physicists. Patients were followed-up clinically and radiologically after SRS to monitor for disease progression and clinical response to SRS. The comparator were patients managed conservatively at diagnosis and followed up clinically and radiologically to monitor for disease progression.

### 2.4. Outcomes

The primary outcome was time-to-disease progression, defined as a tumor volume increase by 25%, according to the RANO criteria [14]. The secondary outcomes included the development of a new neurological deficit or symptom attributable to the meningioma in the active surveillance cohort or SRS in the treated group, the development or increase of peri-tumoral signal change indicative of edema, and subsequent need for an intervention or re-intervention in both groups. In the SRS group, the incidence of secondary malignancy was also evaluated.

### 2.5. Statistical Analysis

Statistical analysis was performed using R v3.5.0 (R Foundation for Statistical Computing, Vienna, Austria. URL https://www.R-project.org/, access date: 3 December 2021) and SPSS v24.0 (Armonk, NY, USA: IBM Corp). Baseline patient and meningioma characteristics were described using number (%), median (interquartile range [IQR]) or mean (standard deviation [SD]) as appropriate and compared between the active surveillance and SRS cohorts. Continuous variables were compared using Student’s *t*-test or Mann-Whitney U test where appropriate. Categorical variables were assessed using Pearson’s χ^2^ test. To control for confounders of treatment outcome, the two cohorts were matched without replacement in a 1:1 ratio using a tolerance level of 10 units for patient age, tumor volume, and duration of follow-up in SPSS. Matching success was determined based on absence of statistically significant differences in the three aforementioned baseline variables. Missing data were not imputed. To test the difference in the primary outcome measure, Kaplan-Meier analysis was utilized. Statistical significance was examined using the log-rank test. To test the difference in secondary outcome measures, a chi-squared test was used.

## 3. Results

### 3.1. Unmatched Population and Meningioma Characteristics

One hundred and twelve patients were included. Their mean age was 58.8 years (SD = 12.8) and 27 (24.1%) were male. Eighty-four patients (75.0%) had SRS while 28 patients (25.0%) underwent active surveillance. Baseline characteristics for the cohort as a whole and stratified by management choice are detailed in Table 1. The median SRS margin dose was 12 Gy (IQR 12–13.5) and the median maximum dose was 25 Gy (IQR 24–28). The median number of isocenters was 9 (IQR 6–11). The median treatment volume was 3.0 cm^3^ (IQR 2.0–5.5). The median clinical follow-up durations following SRS and in the active surveillance cohort were 44.0 months (IQR 24.0–72.0) and 42.0 months (IQR 21.8–66.0), respectively (*p* = 0.547). The median duration of neuroimaging follow-up in the SRS cohort was 36.0 months (IQR 18.0–84.0) compared to 42.0 months (IQR 21.8–66.0) in the active surveillance cohort (*p* = 0.659).

### 3.2. Matched Population and Meningioma Characteristics

Twenty-five patients remained in each cohort after matching. Comparisons of baseline characteristics across the two matched cohorts are provided in Table 2. The median SRS margin dose was 12 Gy (IQR 12–12.5) and the median maximum dose was 24 Gy (IQR 24–28). The median number of isocenters was 7 (IQR 6–11). The median treatment volume was 3.0 cm^3^ (IQR 1.0–7.0). The median clinical follow-up durations following SRS and in the active surveillance cohort were 38.0 months (IQR 26.0–63.0) and 42.0 months (IQR 27.0–66.0), respectively (*p* = 0.641). The median duration of neuroimaging follow-up in the SRS cohort was 36.0 months (IQR 25.0–60.0) compared to 42.0 months (IQR 27.0–66.0) in the active surveillance cohort (*p* = 0.585).

### 3.3. Radiologic and Clinical Outcomes in the Unmatched Cohorts

In the unmatched cohorts, radiological tumor progression occurred in 13 patients (46.4%) managed with active surveillance. None of the patients treated with SRS had disease progression (Figure 1, *p* < 0.001). No new attributable symptoms were observed in the active surveillance cohort. Of the SRS cohort, five patients (6.0%) had new symptoms attributable to treatment after a median of 6 months (IQR 2.5–12.5) (*p* = 0.329). Symptoms were headache (*n* = 3), headache and blurred vision (*n* = 1) and seizure (*n* = 1). Imaging for these patients demonstrated peri-tumoral signal change indicative of edema due to inflammatory radiation effect. Treatment with corticosteroids was required in all cases and additionally, an anti-epileptic in one case. Symptoms were resolved in all cases by the last follow-up 18 months (IQR 3.0–46.5) after treatment. Four additional patients (4.8%) developed asymptomatic peri-tumoral signal change evident on MRI after a median follow-up of 8.3 months (IQR 6.0–10.5), compared to one patient (3.6%) who underwent active monitoring, after 42 months of diagnosis (*p* = 0.792). Treatment with corticosteroids was not deemed necessary. No other cases of developing/worsening peri-tumoral signal change were observed in the active monitoring cohort. There were no cases of secondary malignancy in the SRS cohort within a median follow-up duration of 36 months. 

### 3.4. Radiological and Clinical Outcomes in the Matched Cohorts

In the matched cohorts, radiological tumor progression occurred in 13 patients (52.0%) managed with active surveillance. None of the patients treated with SRS had disease progression (Figure 2, *p* < 0.001). No new attributable symptoms were observed in the active surveillance or SRS cohorts. One patient treated with SRS (4.0%) developed asymptomatic peri-tumoral signal change evident on MRI after 13 months of follow-up compared to one patient actively monitored, 42 months following diagnosis (*p* = 1.000). Treatment with corticosteroids was not deemed necessary.

### 3.5. Need for Surgery or Further Radiation Therapy

Three patients managed with active surveillance (10.7%, unmatched; 12.0%, matched) underwent an intervention for tumor growth after a mean of 30 months (SD = 20.8). Two patients underwent gross total resection of WHO grade 1 meningiomas with no recurrence after 12 and 19 months of follow-up. No medical or surgical complications occurred. One patient underwent fractionated stereotactic radiotherapy for meningioma growth with no further progression 33 months after treatment. The patient had new headaches after treatment with no evidence of peri-tumoral signal change on imaging. Treatment with simple analgesia was sufficient. No patients subject to SRS underwent surgery or further radiotherapy.

## 4. Discussion

In this sub-analysis of the IMPASSE study, upfront treatment with stereotactic radiosurgery reduced the risk of radiological progression of incidental frontobasal meningiomas. In the matched cohorts, the risks of symptomatic, peri-tumoral edema requiring corticosteroid treatment as a result of SRS and symptom development due to meningioma growth in patients managed with active surveillance were similarly low.

Frontobasal meningiomas constitute approximately 6% of incidental meningiomas [9]. They are unique in their lack of *NF2* mutations and, instead, frequently harbor mutations in *AKT1*, *KLF4*, and *SMO* [7]. These driver mutations in frontobasal meningiomas are associated with a lower risk of recurrence after surgery and may also lessen their growth potential. Overall, skull base meningiomas were shown to grow more slowly than non-skull base meningiomas; in a meta-analysis of seven studies, the absolute growth rate of skull base meningiomas was 0.42 cm^3^/year less than that of non-skull base meningiomas [15]. In a retrospective study of 113 patients, only 15 (39.5%) of 38 skull base meningiomas showed growth, whereas 56 (74.7%) of 75 non-skull base meningiomas showed growth [16]. No studies examined the growth characteristics of frontobasal meningiomas in isolation. In this study population, approximately 50% of conservatively managed frontobasal meningiomas demonstrated growth; however, this did not result in symptomatic progression in any of the patients. Retrospective studies of SRS in 41 patients with an olfactory groove meningioma [14], and 763 patients with tuberculum sellae meningiomas [11], reported a 95% local control rate and new symptoms arising in 7–10% of patients. However, these studies included a mix of asymptomatic and symptomatic patients treated previously with surgery. In this study, SRS stopped growth in all patients, and this was not accompanied by symptomatic edema in any SRS cases of the matched cohorts.

The operative outcomes of symptomatic skull base meningiomas and how they compared to outcomes of SRS led authors to recommend early prophylactic treatment with SRS. In a study of 562 surgically managed skull-base meningiomas, 21% of the patients experienced neurological deterioration after surgery and the 30-day mortality rate was 2% [8]. Similarly, a study of 294 operated meningioma demonstrated patients with a skull base location had a worse performance status at 1 year after surgery than patients with a convexity meningioma [17]. However, offering SRS at diagnosis presumes a need for surgery in all patients with an incidental frontobasal meningioma. In this study, about 90% of patients managed with active surveillance remained intervention-free within 5 years of diagnosis and all patients remained symptom free following their initial conservative management strategy. This is similar to outcomes of patients with an incidental meningioma managed conservatively in other studies [4,5,6]. 

On balance, these data would suggest that either SRS or active monitoring are appropriate for the management of patients with an incidental frontobasal meningioma. However, factors such as the economic impact of treating all patients with SRS, patient anxiety during active surveillance, and patient preference, should be considered and may prove the decisive factors in choosing between the two management strategies.

### Study Strengths and Limitations

The study has several strengths including the relatively large size of the study population. Participation from 14 centers across various countries improved the generalizability of the study findings and added to its strengths. The study also has limitations. The retrospective nature of the study design is inherently biased by patient and treatment selection. Decisions for SRS and active surveillance could not be decided based on the collected data. A breakdown of a frontobasal location was not available and subsequent stratification of outcome was not possible. Not all included patients had a pathological diagnosis, and, thus, potential inclusion of patients with WHO grades 2 and 3 meningiomas may have confounded the results—although this is unlikely given the slow growth rates in both cohorts. The median follow-up duration was approximately 4 years. Longer surveillance is needed to determine the long-term risk of tumor growth and associated neurological deficits. Assessments of quality of life and cost of care were not feasible, and both are important factors in decision making. RANO criteria were applied at each participating center however there was no centralized imaging review. Therefore, inter-, and intra-rater reliability could not be assessed. Surveillance protocols for radiological and clinical assessment were dependent on institutional practices. Beyond traditional SRS parameters including dose and tumor volume, we did not collect specific data on variations in SRS techniques or devices utilized. Pseudoprogression as an endpoint of SRS was not collected. 

## 5. Conclusions

Active surveillance and SRS are acceptable management options for patients with an incidental frontobasal meningioma. The economic impact of treating all patients with SRS upon the healthcare system, patient anxiety and patient preference should be considered when counselling patients about treatment options.

## Figures and Tables

**Figure 1 cancers-14-01300-f001:**
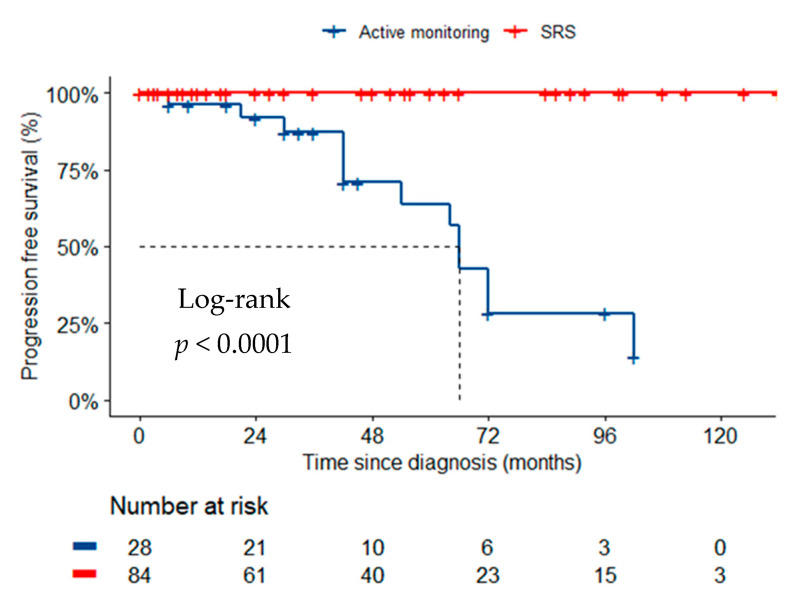
Kaplan Meier curve of progression free survival across the unmatched two management groups: SRS and active monitoring.

**Figure 2 cancers-14-01300-f002:**
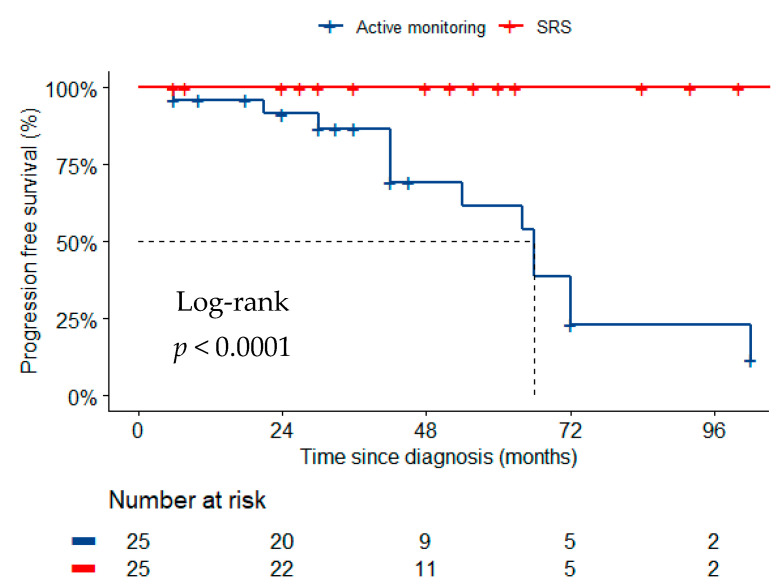
Kaplan Meier curve of progression free survival across the matched two management groups: SRS and active monitoring.

**Table 1 cancers-14-01300-t001:** Baseline characteristics for the whole population and for the unmatched SRS and active surveillance cohorts.

Baseline Characteristic	Total (*n* = 112)	SRS (*n* = 84)	Active Surveillance (*n* = 28)	*p*
Age (years), mean (SD)	58.8 (12.8)	57.6 (12.5)	62.1 (13.3)	0.113
Sex, N (%)				0.251
Male	27 (24.1)	18 (21.4)	9 (32.1)	
Female	85 (75.9)	66 (78.6)	19 (67.9)	
KPS, median (IQR)	90 (90–100)	90 (90–100)	95 (75–100)	0.754
Meningioma volume (cm^3^), median (IQR)	2.0 (1.0–4.0)	2.0 (1.0–4)	1.7 (0.9–3.1)	0.137
Laterality, N (%)				0.505
Right	29 (25.9)	23 (27.4)	6 (21.4)	
Left	42 (37.5)	30 (35.7)	12 (42.9)	
Midline	29 (25.9)	19 (22.6)	10 (35.7)	
Missing	12 (10.7)	12 (14.3)	0 (0)	

**Table 2 cancers-14-01300-t002:** Comparison of baseline characteristics between the matched SRS and active surveillance cohorts.

Baseline Characteristic	SRS (*n* = 25)	Active Surveillance (*n* = 25)	*p*
Age (years), mean (SD)	59.7 (9.9)	60.8 (11.3)	0.702
Sex, N (%)			0.185
Male	4 (16.0)	8 (32.0)	
Female	21 (84.0)	17 (68.0)	
KPS, median (IQR)	90 (90–95)	100 (80–100)	0.405
Meningioma volume (cm^3^), median (IQR)	2.0 (1.0–5.5)	1.7 (0.9–2.9)	0.272
Laterality, N (%)			0.424
Right	7 (33.3)	5 (20.0)	
Left	9 (42.9)	10 (40.0)	
Midline	5 (23.8)	10 (40.0)	
Missing	4 (16.0)	0 (0)	

## Data Availability

The data presented in this study are available on request from the corresponding author.

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
