# Peer review of "Comparison of Active Surveillance to Stereotactic Radiosurgery for the Management of Patients with an Incidental Frontobasal Meningioma—A Sub-Analysis of the IMPASSE Study"

_cancers, 2022, doi:10.3390/cancers14051300_

Round 1

Reviewer 1 Report

This an interesting and important report, although retrospective. It is well written. On the line 228: in any SRS cases the matches cohorts, should be of the matches cohorts, I think.

Author Response

Thank you for your review and feedback. The sentence has been reviewed and adjusted as recommended. 

Reviewer 2 Report

This retrospective multi-center study investigate the outcomes of patients with an incidental  meningioma in a anterior cranial fossa, managed with active surveillance (n:28) compared  to stereotactic radiosurgery (n:84).

In a 5 years follow up  SRS improved the radiological control  but the risk of symptom development was similarly low in both groups. Author conclude that both strategies may be acceptable as treatment options  focusing on the economic impact of SRS on healthcare system, patient anxiety and patient preference  in the final therapeutic decision.

The paper is interesting and well written. It adds something new to the Literature. 

As Authors explained in the section "Limitations" a not clear pathological diagnosis of the cases studied is a bias , as a limited follow up to clarify better the role of these two therapeutic strategies .

A more extensive revision on Literature should be performed, analyzing data of this multicentrica study ion the basis of the existing ones. 

Author Response

Thank you very much for your feedback and comments. 

With regards to the absence of a pathological diagnosis, the absence of any patients succumbing to their disease or requiring an intervention revealing other pathologies, it makes the likelihood of a higher grade meningioma or metastasis highly unlikely.

With regards to the discussion, we have aimed to discuss the salient points arising from the study in the context of existing literature without much waffle. The management of incidental meningiomas on the whole and risk factors for progression has been studies extensively and is felt to be outside the scope of the study question. 

Reviewer 3 Report

In this study, the authors compared the outcomes of patients with an incidental frontobasal meningioma who underwent SRS or active surveillance (median FU 4 years). They concluded that SRS improves tumor control despite a small risk of development new symptoms related to SRS treatment. I have recently read an analogous paper published by the same authors entitled "stereotactic radiosurgery versus active surveillance for asymptomatic, skull‐based meningiomas: an international, multicenter matched cohort study". I would suppose that the 112 patients with frontobasal meningioma (olfactory groove, planum and tubercular sella meningiomas) were already included in this study. Therefore, I do not see something new or the novelty. However, the manuscript is well written. The authors reported that radiological tumor progression occurred in 13 patients managed with active surveillance and 3 of them underwent treatment. What about the remaining 10 patients?

Author Response

Thank you very much for your feedback and comments.

Agreed that the this cohort was included within the overarching skull base manuscript published recently however it was felt that this cohort is unique in its location and also molecular markers that may be harboured by the meningiomas.

The patients who did not receive an intervention remained under follow-up as there no clinical symptoms or signs.

Reviewer 4 Report

The article has been well-written and offers a good view about management of a specific class of meningiomas. Perhaps I think it would have been better to exclude tuberculum sellae meningiomas, potentially able to create early visual impairment, not reversible by radiosurgery and so to be treated early. The work contributes anyway so much to a very interesting subject. 

Author Response

Thank you very much for your feedback.

Unfortunately when creating the database, there was an oversight and frontobasal meningiomas were grouped together from the onset without stratification. Therefore splitting them into different subgroups was not feasible. Moreover, it was felt that these meningiomas share similar molecular and clinical characteristics in terms of growth making the grouping valid. Interestingly, visual impairment was not a feature of the cohort managed conservatively and this cohort did include tuberculum sellae meningiomas. 

Reviewer 5 Report

Thank you for let me reading the article. The conclusion of using SRS early in the frontobasal meningioma is plausible. But the title of this manuscript to too exaggerating and title shows already results. it should be shortened to eg. "comparison of survaillance vs SRS". Otherwise the finding is not so cutting edge but easy to understand.

Author Response

Many thanks for your comments and feedback. The title of the study has been adjusted as recommended.